# A system analysis of the mental health services in Norway and its availability to women with female genital mutilation

**Inger-Lise Lien** [ID]*, **Cecilie Knagenhjelm Hertzberg**¤

Norwegian Centre for Violence and Traumatic Stress Studies, Oslo, Norway

¤ Current address: Centre for Medical Ethics, Institute of Health and Society, University of Oslo, Oslo, Norway

* i.l.lien@nkvts.no

## Abstract

### Background

This article explores mental health services in Norway and their availability for women subjected to female genital mutilation/cutting (FGM/C). The article focus on the system of communication and referrals from the perspective of health workers, and aims to identify bottlenecks in the system, what and where they are to be found, and analyze how different mental health services deal with Sub Saharan African (SSA) women in general, but in particular with respect to FGM/C.

### Method

The study was conducted in Oslo, Norway, using a qualitative fieldwork research design, with the use of purposeful sampling, and a semi-structural guideline. One hundred interviews were done with general practitioners (GPs), gynecologists, psychologists, psychiatrists, midwives and nurses.

### Analysis

A system analysis is applied using socio-cybernetics as a tool to identify the flow of communication and referrals of patients.

### Findings

The study shows that borders of subsystems, silencing mechanisms, regulations and "attitudes" of the system can lead to women with SSA background having difficulty getting access to the specialist services. High standards for referral letters, waiting lists, out pushing to the lower levels, insecurities around treatment and deference rules silencing mental health issues during consultancies, have a negative impact on the accessibility of services. Consequences are that mental health problems due to FGM/C are under-investigated, under-referred, and under-treated and a silenced problem within the mental health services for women.

**Data Availability Statement:** All relevant data are within the manuscript.

**Funding:** The Norwegian Research Council has financed the main part of the research. NKVTS with support from the Ministry of Justice has

contributed to the financing of the research. No -
The funders had no role in study design, data
collection and analysis, decision to publish, or
preparation of the manuscript.

**Competing interests:** The authors have declared
that no competing interests exist.

## Conclusion

A better integration of subsystems at the specialist level with the GP scheme is necessary, as well as providing competence on FGM/C to the different levels. It is also important to strengthen and integrating the services at the Municipal level and provide information to SSA women about the low threshold services.

## Background

This article is based on a study "Physical and psychological health care for girls and women with Female Genital Mutilation/Cutting (FGM/C)", financed by the Norwegian Research Council. The study has several modules that will provide basis for a number of publications dealing with different aspects of health care. In this particular article we will address the availability and accessibility of mental health care services. The study is based on interviews with health workers who reflect on the system and the way Sub Saharan African women (SSA women), who potentially suffer from mental health consequences of FGM/C, or from other mental health problems, are received by the system.

In Oslo, where this study took place, immigrants comprise 33.1% of the population, and 17.8% has an African background, including 15,608 Somali and 7,208 of Ethiopian and Eritrean descent [1]. In 2013, the number of women and children originating from countries that practice FGM/C was estimated to constitute around 2% (44,467) of the total Norwegian female population. Around 17,300 girls and women (0.7% of the total population) are estimated to have already been subjected to FGM/C prior to migration to Norway [2]. In this study we were particularly interested in the Sub Saharan African (SSA) women within the healthcare services (i.e., Somali, Ethiopian/Eritrean and Gambian women) who come from countries where FGM/C is practiced.

The World Health Organization [3] has distinguished between four main types of FGM/C: (1) clitoridectomy, which involves partial or total removal of the clitoris and/or prepuce; (2) excision, which involves partial or total removal of the labia minora and/or the labia majora; (3) infibulation, which involves narrowing the vaginal opening through the creation of a covering seal with or without removal of the clitoris; and (4) others, which encompasses all other harmful procedures done to female genitalia for non-medical reasons.

A series of systematic reviews of health consequences of FGM/C was conducted by the Norwegian Knowledge Group for Health Services, finding that FGM/C leads to pain, excessive bleeding, problems with wound healing and complications that include clitoral inclusion cysts and urinary problems [4]. One of the systematic reviews included psychological and sexual consequences [5], finding that women with FGM/C experienced increased pain and reduced desire and pleasure during sex. They were more likely to experience psychological disturbances; suffer from anxiety, somatization, phobia and low self-esteem; and have a psychiatric diagnosis. Newer studies [6–8] have confirmed findings that women with FGM/C suffer from higher prevalence of symptoms of severe depression. A study from the Netherlands [8] found that 1 of 6 of their African female informants had indications of PTSD and that 1/3 of the 66 informants, showed symptoms of depression and anxiety as well as chronic psychosocial problems, that the researcher argued, seemed to be linked to the early experience of FGM/C in their childhood. However, it is not clear that women themselves would link their present depression or traumas to the FGM/C procedure that may have taken place in the past [9]. In most cases women can very well remember the pain they went through, but as FGM/C is

usually seen as something necessary and good, even though very painful, being cut would be categorized as a normal health condition for any woman, and linking a depressed state of mind to the procedure performed in childhood, would thereby be difficult to make. According to Berg et al. [4] the continuation of FGM/C is among other factors, attributed to the fact that the women themselves see the procedure to give health benefits, rather than negative health consequences. However, several researchers have found that PTSD is one of the psychological sufferings that can be linked to FGM/C [5, 8, 10].

A literature review of immigrants' mental health in Norway [11] concluded that they had higher levels of mental health problems in general, such as depressive symptoms and emotional problems compared to the native population. Other studies have also found mental problems to be more prevalent in refugees, especially immigrant women who have experienced negative life events and extreme traumatic stress such as war, violence and torture [12–14]. Psychological problems were found to be consistently higher among adolescent girls than boys and among adult women than adult men [15]. A study of life satisfaction among immigrants in Norway [16], found that the prevalence of mental problems in immigrants was twice that of the general population. On the other hand, immigrants from Somalia and Eritrea, according to this study, had more or less the same degree of psychological problems as the general population, but one in five Iraqi and Iranian immigrants had psychological problems. As Somalis are integrated in the labor market to a lesser degree than the general population and may also have war experiences and suffer from poverty, one should expect their mental health problems to be higher. The positive results among Somalis and Eritreans are described as a mystery by the researcher, conducting the study, who thinks that the cause may be that the informants compare their present situation to the hardships and war experiences they suffered through in their homeland [17]; thereby tending to underreport their present difficulties. However, none of the Norwegian studies mentioned above have focused on the impact of FGM/C on mental health.

It has been documented that patients in Norway with non-western backgrounds often go to emergency services (ER) instead of arranging a consultation with a GP [18]. There are therefore differences in the use of the ER services and GPs between different groups of SSA immigrants. Immigrants from Somalia use ER services more often than Ethiopian, Eritrean and Gambian immigrants [18]. A study from Norway, [19] found that Somali refugee women in Oslo lacked the capacity to obtain, understand and act upon health information and services, to make appropriate health decisions. Ververda and Ismail [20] found that Somalis did not have a word for depression, making it difficult for them to talk about their feelings, especially because feeling bad was often a taboo subject. In an analysis of trust in the health services in Norway, Næss [21] argued that "cramped timeslots and a 'cut-to-the-case- attitude' among doctors" can be a poor foundation for trust in the health system." Næss maintains that Somalis compensate for their lack of trust and health capital by using bridge-builders within their own ethnic groups for advice when using the system [21].

A systematic review [22] of qualitative studies of health care providers' perspectives on health care for FGM/C survivors, has identified 20 descriptive themes within 28 papers analyzing health care. The review, which covers health care within several countries, concludes that there are communication and language problems during consultancies, as well as lack of competence, the FGM/C topic is taboo and silenced, there is a lack of trust and difficulties identifying FGM/C. The review concludes that there are needs for guidelines, care pathways, care models and protocols. The studies that the review has analyzed, have mostly focused on physical health, rather than psychological health. Psychological problems were mentioned mostly by nurses and midwives according to the reviewers that ask for studies of pathways in the system, and for more holistic models and analysis of barriers in the systems. In our study the aim has

been to fill in some of the gaps mentioned, by looking at mental health care specifically, rather than physical health care, as seen from a system perspective.

In 2001, Norway implemented the General Practitioner Scheme, which assigned general practitioners (GPs) as the gateway to medical services [23]. This system consists of three core elements [24]: (a). It is a list-system that registers and links all inhabitants to a personal GP. (b). The responsibility for the patient is placed on a named medical doctor, so there is an address for further follow ups after consultations with second-line services. (c). The system has a contract with the municipality, so the GP has certain duties to fulfill for the municipality.

Since the start of the scheme, the number of GPs has increased by 30% [25]. However, the volume of patients has also increased due to immigration and a growing elderly population. The number of yearly consultations per person has increased from 2.5 consultations in 2006 to 2.7 in 2016 [25]. Studies have indicated that 35% of GPs work more than 62.2 hours a week, while the average workload is 55.6 hours a week, an average increase of 7 hours since 2014 [26]. There is a general discussion in the media about a crisis in the GP scheme because of overload, burnout and long waiting lists in the system.

This article has several purposes. First, it describes the organization of the GP-scheme and its relation to mental health and FGM/C in Norway, with Oslo as the illustrating case, focusing on the health system as a socio-cybernetic system with subsystems. Next, it describes the communicative dynamics of the system in the form of written referrals to the specialist services, including related challenges and feedback loops. Consequently, it gives a description of boundary maintenance of the subsystems with the effect of protecting the system through standard requirements for inclusion of patients and practices of rejections. This work then identifies where we can find easier entries for SSA women who may be cut and suffer mental health problems.

The overall aim of the analysis is to give insights into borders within the mental health care system in order to understand what happens, and where and when SSA women are seeking health services for mental help due to FGM/C. The intention is to provide this knowledge so that it can be used to improve access to health care and treatment for SSA women in general, but in particular for those who suffer mental health problems related to the practice of FGM/C.

## The health system as a socio-cybernetic system

In order to understand the working of the mental health system in relation to the SSA women, we will use a socio-cybernetic approach that represents a theoretical perspective in the science of systems and organizations [27]. Socio-cybernetics explore regulatory systems—their structures, constraints, and possibilities. Key concepts are self-regulation, control mechanisms, information and communication [28]. The analysis looks for viability or for equilibrium of the system, or homeostasis, and systems are often described as circles that operate in an environment [29]. Overload, or overheating, are also concepts that are useful analytical tools. A car can be used as a metaphor in which speed and heat lead to overload, or to overheating, so that the car engine eventually loses energy and stops [30]. This metaphor is an example of reduced energy, or entropy, where equilibrium is broken. A change of entropy means an increase in disorder and a lack of energy to perform the work in the system.

According to the sociologist Niklas Luhmann [31], a social system can be described as a communicative system that has closure. It is autopoietic: it creates itself from itself and maintain itself by being closed. Luhmann holds that the system acquires freedom and autonomy of self-regulation through *indifference* to its environment. However, the environment can have an impact on the system that is mostly described by the concept of *disturbance* that must be dealt with through an *internal modification* [32].

Thus, *disturbances of the system* can lead to a new setting of the thermometer as has been mentioned by Bateson [33], which can be used to cool off or to heat up the system. "The thermometer" of the system constitute what Bateson calls: "the attitude of the system" [33]. The thermometer is made of attitudes of care and respect and standards for entry into the institutions, and can be set high or low, and thereby regulate care and movement of patients in the health system. Socio-cybernetics that deal with the flow of energy and communication can serve as useful analytical tools when studying the health care system, and the flow of patients within it. The benefit of using a socio cybernetic analysis is that it provides tools for understanding how complex systems work. By following the flow of energy, people or patients, through the different parts of the system and investigate how parts interact with each other, it is possible to grasp how change in one part of the system can have an effect on the other. It can also show how different flows can be directed and redirected when there is blockings, overflow, or slack in the system. A health system is complex and difficult to understand and study, and it is easy to become judgmental. Cybernetics can be a way into analyzing such a system neutrally, especially when change is taking place. A socio-cybernetic analytical model also provides a way of describing a complicated system with the flow of communication and personnel between levels in a non-moralistic way.

In the 70s, Stafford Beer [34] built a cybernetic model for health care, the Viable System Model, for the Ontario government and studied the balance between the pool of people who are well and those who are ill, to retain the category of the ill to that of the well. The health system can be described as a social system that is anchored by the dichotomy of well and ill, while the system of law is anchored by the dichotomy of legal and illegal. Cybernetic models are used in psychology as well as in biology and other behavioral sciences [35]. This perspective has its focus on relationships that are dynamic. Feedback is one of the core concepts and relates to information and communication in the system.

## Method

This study is based on qualitative data collection in Oslo by two social anthropologists. From December 2017 to December 2018, more than one hundred interviews were done using a semi-structured interview guide (see S1 Appendix). The interviews lasted for approximately 1–2 hours, and some informants were interviewed twice (S1 Table).

Getting access to GP informants proved difficult, due to the heavy work load that they have. It forced us to start out with our own networks of acquainted GPs, which led to a methodological snowball effect [36], as well as a purposeful sampling procedure [37]. We sought out informants representing maximal variations when it comes to established institutions, organizational levels, and professions involved in Oslo. As there are more women than men within health care, women were in majority, while men (N = 6) were in a small minority. We performed many of our interviews in the workplaces of our informants in order to base our findings, not only on what was said, but in addition on what we observed. The institutions at the municipal level (Fig 1) were more easily accessible than the institutions at the specialist level, like the Regional Psychiatric Centers. We visited and obtained information from Ullevål Hospital and Rikshospital, and four Regional Psychiatric Centers (DPS). We also obtained information from nine Rapid Psychic Help (RPH); visited six Healthy Life Centers (HLC); four Activity Houses, and three Health Centers. Furthermore, we interviewed two translators and three bureaucrats working within Oslo's health administration. We followed an interview guide (S1 Appendix), and based on this, we talked very openly with the health workers and allowed topics to be discussed freely. Referral practices, communication and relationships with other institutions were topics that proved to be a challenging issue for the health workers. We soon discovered that mental health

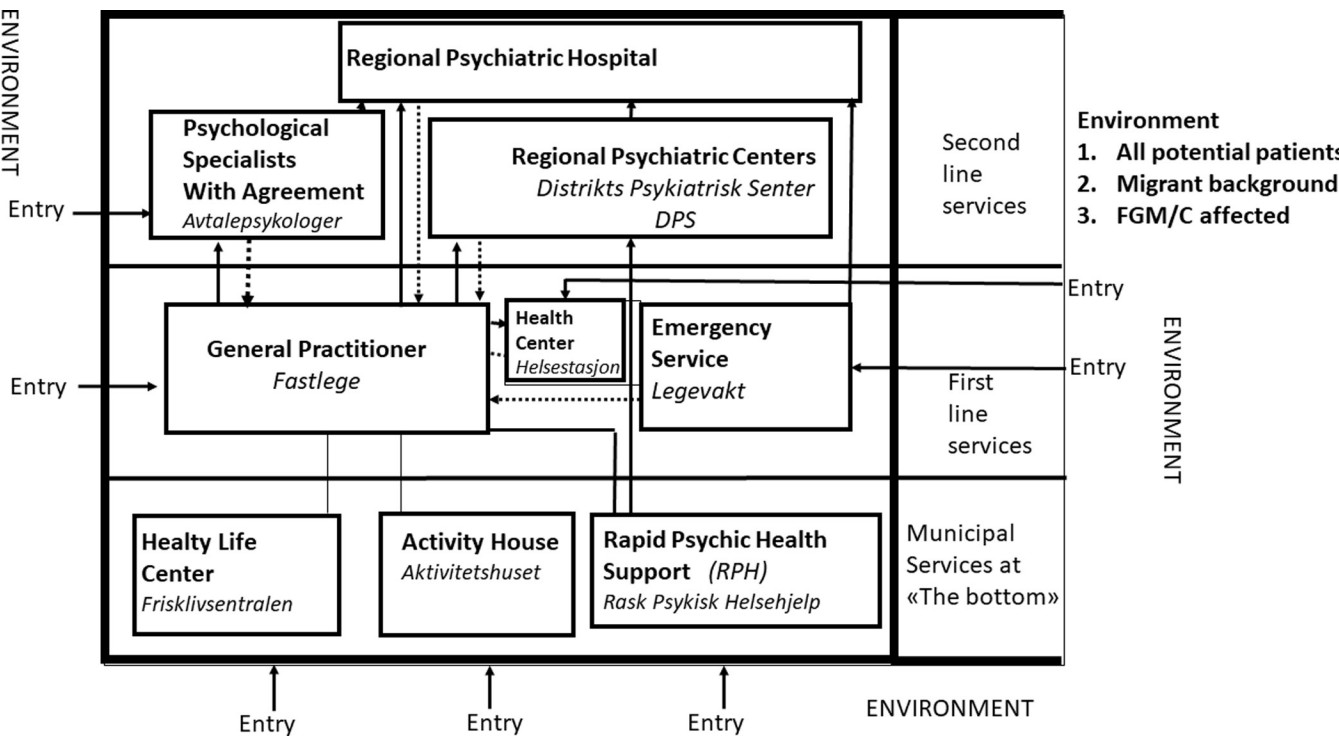

**Fig 1.**

issues due to FGM/C was a topic that very seldom came up in GPs consultancies with SSA women, leading us to broaden the discussion and ask about mental health in general that GPs had been dealing with in consultancies with SSA women. Of the researchers, one were in her sixties and the other in her early thirties, both of them having done fieldwork in different parts of the world and having cultural experience and training. The researchers took a humble and respectful role during their interaction with health workers. When health workers described cases they were asked to present the cases as neutral and as anonymously as they could. Qualitative anthropological data collection implies validating the data as the interviews proceed by checking information when interviewing new informants. In addition, three focus groups were used to validate information. Descriptions of the system that interviewers presented, were analyzed during the data gathering process by the use of socio-cybernetic theory.

The interviewed health workers have been given written and oral information about the project. Prior to the interviews, informants have signed a declaration of consent. We did not ask health workers any personal questions other than age and gender. Data has been presented in the manuscript and analyzed anonymously. The project got ethical clearance from the Regional Committee for Medical and Health Research Ethics in Norway, which granted permission the 12.09.2017. All necessary permits were obtained for the described study, which complied with all relevant regulations.

## Findings and discussion

### A system of mental health and its borders and entries

Above is the model of the health system that targets mental health. This system is a part of the health system that women can enter who suffer from mental health problems in general or the psychological consequences of FGM/C.

Any healthcare system is complicated and contains many subsystems. Some subsystems are governmental, while others are run by the municipality, and other subsystems are private but receive some economic support from the government. These systems communicate increasingly through the Internet [25], but consultations are usually face-to-face encounters. In the socio-cybernetic health model that we have drawn, the environment is something that is outside the system but becomes relevant to the system when individuals enter the "unwell" category. In the environment there are new groups of potential patients arriving from foreign countries who are well or unwell and may require healthcare because of their symptoms. Motivated by these sensations, they may contact health workers in the system to obtain knowledge, diagnosis and treatment.

The system used to have only two levels, the first level where the GP has been the hub, and the second level containing the specialist institutions. The GP has been given the juridical obligation to refer patients further to the second line if he considers that the patient has a need. This referral must be assessed by the institution that is going to receive the patient. The patient has the right to get the GP referral assessed by the institutions at the specialist level, like the Regional Psychiatric Service (DPS), but it is on the basis of the assessment of diagnosis and priority done by the DPS or the Psychologists with agreement, that the patient will be given a right for treatment, not the GPs referral itself. In recent years the mental health services have been overloaded with long waiting lists. Thereby the Government has strengthened access to mental health services by adding a new set of low threshold institutions at the municipal level. These low threshold services do not demand that the patient must go through a GP in order to get a referral letter. The low threshold services are free of charge and easily available, but unfortunately some institutions at this level have got waiting lists that will delay entry to the services. The new institutions like the Rapid Psychic Health Support and the Activity Houses have psychologists employed as well as other professions. The solid lines in our model indicate the referral lines from the GP to psychological specialists with agreement (Avtalepsykologer), to psychiatric hospitals and to regional psychiatric centers (DPS) at the specialist level, and the stippled lines represent the feedback tracks from these institutions to the GP via discharge summaries, journals and other notes; these are mostly sent via the Internet.

The environment is the frame of the system and is not static but can change due to environmental disturbances such as epidemics, new patient groups, immigration, and many other factors. The employed health workers within the system have gained some theoretical knowledge about patients with FGM/C, but perhaps not enough practical knowledge to adequately help these patients [9]. Within the first line services there are two main entries, either through a GP or through emergency services (ER services), but pregnant women can take priority for follow ups and registration with health centers (Helsestasjon). The health centers are particularly directed towards preventive health work for children, youth and pregnant women, and are usually employed by midwives and nurses.

The way to enter the low threshold services is by calling them, or to meet up directly, but also through a recommendation by the GP who provide an address written on a piece of paper. The added subsystems at the municipality level are meant to be preventive in the sense that they will treat mild illnesses. As these municipal services are low threshold services that are open for all potential patients, they require health capital [38], familiarity and trust [21], as well as initiative and competence about the system.

## Sub systems at the first level

**The emergency.**   The emergency services (ER) are intended for patients when the GP system is closed and there is an acute need for medical assistance. However, this system is often

used by patients as a replacement for a GP when patients choose to not wait for an appointment or are not registered with a GP.

During interviews, the medical doctors at the ER services, distinguished between a GP assessment and an emergency assessment that only concentrates on one particular health issue at a time and does not involve follow-ups. When entering the system in this way, patients will not adequately be part of the same feed-back system that the GP is part of; instead, they will be treated for an acute health problem, such as a sudden ear or throat infection, and not lead to any follow-ups.

When we interviewed medical doctors in the ER, they could not remember occasions where a patient had visited them with concerns related to FGM/C, and they could not remember referring a patient to any psychologist due to psychological problems linked to FGM/C.

**The hub: The general practitioner.** As this is a system analysis, our main focus is on the relationships between the subsystems, feedback system and borders; however, due to the gate-keeping role of the GP, this position is a focal point within the analysis. We begin by looking more closely at encounters between GPs and patients. The interaction between a patient and a GP in a consultancy meeting is influenced by the number of patients that a GP has.

A GP can have a maximum of 2,500 patients on his or her list, but on average, a GP has around 1,200 patients [23]. The time allocated to each patient can be as brief as only 10 minutes per patient, but some GPs have organized their time to allow for 20 minutes for appointments. The lack of health literacy and health capital among patients may require the GP to provide more detailed explanations and use more time than is available to teach the patient about issues that in other circumstances would not be necessary [25]. Health illiteracy, or lack of health capital, is present in patients who have a reduced ability to understand and act upon medical advice. However, there are other restrictions that can impact communication between a patient and a GP. These include norms, values and attitudes as well as rules and acts of deference [39]. Ervin Goffman described how doctors, patients and nurses in a hospital ward used different activities to show appreciation of themselves to a recipient (patients and other health workers) as well as appreciation of the recipient [39]. Acts of deference on the part of a healthcare professional can involve following rules to maintain privacy and respect the patient; in turn, the patient will show respect for the GP. In this study, several GPs communicated that patients show deference in ways that the GPs are not accustomed to, and these acts of deference consume time during an appointment. For example, some SSA women have many layers of clothing and are reluctant to uncover themselves. "This undressing takes half the time that I have available." (GP, male, age 50). Some SSA women are not even willing to show naked skin to the GP or to let the GP touch their skin in order to show deference to the GP as a male. Other women takes their husband with them for translation which can make conversation about private issues difficult. These acts are due to values of deference, but from a GP perspective the acts can be seen as time consuming, irrelevant and disturbing during a medical appointment. The GP often responds by showing deference back: "When there is an African patient, it comes to my mind that she may be cut. But I do not ask. I will not destroy the patient-doctor relationship" (GP, female, age 47). Many of the GP informants came with the same arguments that they do not want to insult the patients' values by talking about sensitive issues, or destroy the doctor-patient relationship: "I think about it when there is an African patient, that she may be cut, but I think that she may not want to talk about it, so I do not ask" (GP, female, age 47).

When respect and sensitivity for culture are shown in this way, health issues may be overlooked, and potentially needed healthcare may not be given. A narrow timeframe inhibit careful questioning and can become an obstacle for elucidating mental health problems that patients suffer from.

Some GPs explained that it is difficult to talk about mental health in general with the group of women they think have been cut. A common complaint was this: "African women do not complain easily. They keep things to themselves. Then it is difficult to help." Cutting is performed in a very private part of the body, and it is invisible from the outside. Complications due to FGM/C needs to be approached respectfully, in the same way that mental health issues, which are often invisible to the observer, need to be carefully unmasked. It require extra time in order to be carefully looked into and discussed. In some GP waiting rooms, we have seen posters on the wall telling patients that they should only talk about one health problem at a time in an effort to reduce the number of medical issues to be addressed during a brief appointment. Less visible or "difficult-to-talk-about" issues like cutting and mental health problems, may become irrelevant and silenced during a consultancy.

Patients themselves may want to keep private issue hidden or private in front of a GP for reasons other than deference. Some GPs have indicated that Somali patients could be afraid of losing their children to the child welfare authorities if a GP suspects that a patient has mental health problems.

When asked about when they should refer a women to a psychologist or a DPS, most GPs answer that on a scale from 1 to 10, where 10 is the highest amount of trouble, they would refer a woman to a DPS for treatment if they think her psychological problems lie between 6 and 7.

None of the interviewed GPs in Oslo had asked a woman about the mental consequences of FGM/C. No female patients had asked them for help to deal with mental health problems due to FGM/C. As a result, FGM/C is an undiscussed issue at GP consultancies, although several GPs who took courses on this topic during their studies, have suspected that some of their patients were cut. Sometimes a GP has coincidentally discovered that a woman has been cut, especially when testing for cancer. According to GPs, very few immigrant women actually request cancer tests, and GPs seldom investigate SSA women's genital areas. Two of the GPs who were educated abroad, were completely unfamiliar with the topic, and were shocked when we, the researchers, told them about the practice of FGM/C. Psychological problems related to FGM/C seem not be a prominent topic addressed by GPs in Oslo, even though many impacted women live there. However, GPs mentioned that mental health problems like depression are a widespread problem that many women with SSA backgrounds suffer from.

If it was obvious to a GP that their patient needed psychological help during a consultation, the GP would first assess the patient using different scales, like a depression scale. If the scale indicated that the patient suffered from depression, the GP would write a referral. If the referral was not approved by DPS or the psychologists with agreement (Avtalepsykologer), the GP would have to decide whether to rewrite and resend it. Dismissals are generally due to referrals that do not meet certain standards. A GP often has to submit a referral a second and a third time, and make a phone call as well, to get his or her patient access to second-line services. A couple of GPs had, on their own initiative, taken courses in cognitive therapy that they offered to their patients after work in order to help them, as it was difficult to get patients accepted by services at the second level.

**Health centers.** Health centers are places where FGM/C naturally must be dealt with. During their first consultation, pregnant women are asked if they are cut and/or need to be opened (deinfibulated). Midwives can perform all the examinations that GPs do, related to pregnancy control. They can also perform the surgical opening procedure (deinfibulation) on a woman who has been sealed (infibulated), or they can send her to the hospital. In the health center, within the most immigrant dense area of Oslo, the number of pregnant Somali women has been stable since 2014. An average of 440 women are pregnant yearly, and of these, 10% (N = 46) have a Somali background, and 1.8% have an Ethiopian/Eritrean background. One of

the midwives reporting seeing five infibulations over 14 years of service, mostly in Somalian women; another midwife had seen two infibulations over two years. In the three boroughs we visited in Oslo, all within dense immigrant areas, there were psychologists employed at the municipal level. However, health centers visited, had not sent a woman to a "low threshold psychologist" due to FGM/C. The midwives did not have referral rights to send women to the DPS or to the Avtalepsykologer. It seems that health centers in municipalities are the places where one can find FGM/C-affected women when they are pregnant and being cared for by nurses and midwives. However, mental health issues due to FGM/C do not seem to have had a central focus in these centers, as the topic does not seem to have come up.

## Subsystems at the second level: Opening and closure

**The subsystem of psychological specialists with agreement (Avtalepsykologer).** The psychological specialists with agreement (Avtalepsykologer) have made an arrangement with the municipality to receive patients from the system; they then receive part of the cost for a consultation that is paid by the municipality. Therefore, they are called *Avtalepsykologer* in Norwegian (Psychologists with an Agreement). Their offices are mostly located in the west of Oslo, where the middle and upper class people live, but not in the eastern part, where the majority of immigrants and, consequently, women potentially suffering from FGM/C live. Most of the GPs interviewed in this study, had more or less given up on efforts to refer their SSA patients to this service. As a result, this subsystem has made itself unavailable for many patients. Location, long waiting lists, therapies lasting for a very long time, and very little feedback and communication with GPs, all create boundaries within the health system, resulting in difficulty admitting patients with an SSA background. All of the GPs that we interviewed complained about the inaccessibility of this particular service: "You must just forget the psychologists with agreement" (GP, male, age 40). "Those psychologists with agreement only want patients similar to themselves. They are around 50 years of age and their patients are 50 years, and the patients are all white" (GP, male, age 69). The GPs also complained about lack of feedback from this service, that they did not hear from them at all, or very seldom: "Sometimes I receive a handwritten letter, or a page torn from a notebook, and I do not understand because the handwriting is so bad. If I try to call, they do not even pick up the phone" (GP, male, age 42).

When it comes to mental health problems due to FGM/C or mental health problems among SSA women in general, the GPs that we interviewed did not remember having referred female patients with this background to the psychologists with agreement. This subsystem may provide very good treatment to many patients that have been granted entry, but it seems to be more or less inaccessible for many new, needy patients referred by GPs. The boundary of the system is maintained by the autonomy of the subsystem, the freedom of the psychologists to choose their own patients according to their interests, the long waiting lists, and the long time therapy that is provided. The scarcity of feedback provided to the GPs is also indicative of a service that is poorly integrated with other systems, and particularly with the GP scheme.

**The regional psychiatric centers, or "Distrikt Psykiatrisk Senter" (DPS).** The regional psychiatric center (DPS) is a subsystem protected by a boundary comprising several referral criteria, waiting-lists as well as the autonomy and freedom that the centers have to take in or dismiss patients due to their own professional assessment, even after having taken them in for a first meeting based on a GP's referral.

The way a referral letter to the DPS is written by a GP may or may not be convincing, and some of our GP informants have argued that they have taken writing courses in order to learn which language and words will lead to an acceptance of their referral letter. They are troubled

by the fact that there are professional words that are more convincing, and that ways of formulating the request impact the acceptance of their patients. The referral standards set by the DPS work to maintain boundaries for admitting patients into the subsystem. A patient must pass two thresholds in order to access the DPS. The first is to convince the GP that there is a need; secondly, the GP again needs to have the competence and writing skills to convince the DPS of his patient's need.

The GPs argue that they often must send two or three referrals before a patient is accepted. As one GP said, "We are educated to become clinicians, and not writers." It can take a GP 1.5 hours to write a referral, and if it is dismissed, he or she has to use additional time to improve the referral letter, which increases the workload of the GP. In addition, the patients may return for consultation several times due to the dismissal of the referral. Dismissals thereby become a heavy workload for GPs. Some GPs argue that the DPS focuses not on patients' problems but on the referral letters written. "I think it is very strange that, after I have made a professional judgment that a patient needs psychological treatment at the DPS, which they, on the basis of my letter, can get to the opposite conclusion". After accepting a patient and taking him/her in for the first consultation meeting, a center can again refuse the patient based on their own professional judgement and disregard the referral letter. "It is not unthinkable that they play around with the waiting list, knowing that there will be another referral letter coming after a month or so. Doing this, they have bought themselves more time, and me more work." (GP, male, age 42).

If they do not refuse the patient, the DPS will have 12 diagnostic meetings before starting therapy. As someone employed at the DPS said: "Many patients are wrongly referred, and they are not able to profit from our treatment. Their troubles can be chronic. We have to show clinical judgment, even if we are in doubt. The GP can have an opinion about what treatment the patient needs, but we decide what the patient needs." (Psychiatrist, male, age 50). "Patients who have been treated many times for the same troubles will not be able to profit from our offer. Then we try to send him or her to a GP". (Psychiatrist, male, age 40)."Three hundred patients must leave during a year in order for 300 new patients to be taken in" (Psychiatrist, male, age 50).

Boundary maintenance within the DPS protects the system from an overflow of patients in a situation where there is an extensive need for psychiatric services in general, but these boundaries impact refugees and migrants in particular. In two of the DPSs where we conducted interviews, they said they had very few SSA patients in their centers. One claimed that 50% of their patients had an immigrant background. Two other DPSs in immigrant-dense areas did not have any patients with SSA backgrounds at the time the interview took place.

The problems that patients with an SSA background suffer from, are called "the top three: anxiety, depression and trauma." "Many of the patients have experienced war, seen family members being killed, have experienced and seen somebody being tortured or been raped."

Overall, FGM/C is not an issue that GPs mention when referring a patient to a DPS, it is not discussed by the patient herself, nor does it seem to be an issue that surfaces during treatment. None of psychologists at the DPSs that we interviewed, had experienced or brought up FGM/C. "We ask open questions. It may happen that FGM/C is not mentioned because the patient does not talk about it." FGM/C is therefore an under-communicated, silenced issue on the part of patients, GPs, and DPSs.

Some psychologists in DPSs indicated that it was more difficult to provide therapy to a patient from an immigrant background than to patient from an ethnic Norwegian group. This issue is due to communication problems and problems of cultural understanding. "If I got a girl who was circumcised to my office, I would feel helpless." The psychologists interviewed also mentioned the extra burden created in talk-based therapy to have a translator in the

room. "When using an interpreter, I am using double the amount of time, and time is limited. Then the patients do not get good treatment" (Psychologist, female, age 52).

The informants in the DPSs told us that when they receive patients, it is important to deal with psychosocial issues first; therefore, they often send their patients to the services within municipalities–to get them into appropriate social groups of some kind. The DPSs also have therapeutic groups, but if they think that patients do not fit into their groups, they will seek out other places to send them. "Then you think that you cannot offer the services to this person because he/she does not fit into the group."

Some of the DPS informants were empathic and understanding of the FGM/C issue, but they felt insecure and somewhat incompetent:

> When you work with someone from another culture, you become careful and wondering. You feel unsafe. Then the therapy will not work properly because you do not have the same sets of references. There is no flow in the communication, and you must use more time to understand. (Psychologist, female, age 35)

> They will not get sufficient benefits from the therapies. (Psychologist, female, age 33)

Thereby, boundaries are kept high both within the services between subsystems and outside the services. These boundaries shield FGM/C topics from professional examination. Needy patients from an SSA background seem to be on the outskirts of the system, either because their problems are chronic, too unclear or not serious enough, or because the referral from their GP is not considered sufficient. Patients are thus sent back to GPs or down to municipal services. Due to a high demand for psychological services in general, there is a pressure towards the system which is met by boundary maintenance to protect the system from over-flow. There is a high standard requirement to the referral letters that the GPs are supposed to write, long waiting lists, and dismissals as well as out pushing to the lower levels. In addition there are insecurities among professionals, especially when it comes to communication with patient groups that suffer from harmful traditions such as FGM/C.

## The municipal services

The GP can also recommend that the patient search for help outside beyond specialist services (i.e., seek out municipal health services that do not require a formal reference from a GP). These low threshold institutions have developed during the last 10–15 years. The establishment of the institutions represents a shift of focus from reparation to prevention, but institutions are also doing a lot of reparation. The institutions include activity houses, or Aktivitetshus; the Healthy Life Center, or Frisklivssentralen; and rapid psychic health support, or Rask Psykisk Helsehjelp (RPH). In addition, there are institutions that the municipality supports economically that offer services similar to those of the Healthy Life Center, like the Health Forum for Women in Oslo, where they have physical exercise groups for immigrant women.

**Rapid psychic health support.** The RPH is the only institution that can send formal referrals to the DPS and get a patient admitted to second line services without going through a GP referral. A psychologist in RPH even argued that the RPH has better access to the DPS, as they are often better referral writers than GPs. The institutions that we have mentioned at this level, are meant to be available to all, without charge and without long waiting lists.

RPH emerged as a pilot project in 2012. It was inspired by Improving Access to Psychological Therapies (IAPT), which originated in England. RPH uses cognitive behavioral therapy as a method and it is a short-time service that people typically visit 8–10 times. The service is for

people with mild forms of anxiety and depression, and the goal is to reduce symptoms and enhance quality of life and function for those receiving help. There is not supposed to be a waiting time, but the service is in high demand, and waiting times range from 2 to 5 weeks. There were almost no patients with an SSA background within the RPH in Oslo at the time of this study. However, one SSA patient was found in one RPH in western Oslo. Most other RPHs served ethnic Norwegians, but some in immigrant dense areas served Pakistani and South Asian patients. One RPH in eastern Oslo employed a Somalian psychologist with the aim of recruiting patients with an SSA background. "There have not been many common places for Somalis where we can promote RPH, so we have been to the mosque and talked to the 'district mothers.' Many have restricted language, so we must use a translator, and that can also be time consuming" (Psychologist RPH, female, age 40). Out of the nine RPHs that we contacted, only one reported that a woman had talked about FGM/C; however, the issue was not addressed initially–it arose when the psychologist was assessing the patient. Consequently, FGM/C was described as a big trauma and a taboo, but it was not treated at any of the RPHs.

**Activity houses.**   Women with an SSA background are not often found at *Activity Houses* (Aktivitetshus). "The Somalis don't dare seek out the Activity Houses–psychological issues are taboo. Immigrants understand bodily pain, but if it is psychological, it has to be bad!" (Activity House therapist, female, age 60). The Activity Houses are organized within the department of psychological health care in the Oslo municipality. The houses are often used by the DPS as a place for their patients to find support when their referral is dismissed, or following treatment. There are persons with immigrant background in these houses–Pakistanis, Vietnamese, South Asians as well as Norwegians–but hardly any with an SSA background. The houses organize different artistic activities like painting groups, music groups and different crafts workshops and social groups. Often dinner is made and served. The goal is to create security and counteract loneliness and depression. These houses are places to meet others and build a social networks.

**The Healthy Life Centers.**   Within the locations of the Healthy Life Centers (Frisklivssentralene) (HCL) we found the groups of women that we were searching for. We visited five Healthy Life Centers in the immigrant dense areas in eastern Oslo as well as the Health Forum for Women, where the majority of users had an SSA background. In the most central area of Oslo there was a Healthy Life Center that had 200 users per year, and 90% of these users were not Norwegian. Furthermore, 60% of the users had an SSA background. The focus of these places is physical activities, healthy eating, smoking cessation and dealing with depression. Activities are organized in groups, and there are separate Somali groups and groups only for women. To enter the Healthy Life Centers, a certificate, not a real referral, from a health worker, but not necessarily a GP, is needed. Individuals are often referred by the DPS after the end of treatment, or if the DPS finds that the person cannot be treated by them and dismisses a referral. The Healthy Life Centers offer a twelve-week program, but they can extend this time. There is no need for a diagnosis. Some of the users have been part of an HLC for 5 years.

> We have a lot of immigrant women; at the time we have 80% women, and 80% of them are from other countries (. . .) The Somali women have a lot of worries. We are working to make the socializing a bit better, but they need to believe in themselves. When I ask them what they like to do, they often respond "nothing". I think we can help them a lot more than people think. At the same time, they are so different each time, but they all have one thing in common: they are holding things back. I always need to dig a little and they are not a group that demands a lot. (HLC therapist, female, age 31)

Through the strategy of workout activities, these services can indirectly resolve mental health issues that have been suppressed. There are courses on pain management, healthy eating

and healthy planning as well as different physical activities. The Somalian groups have 20 to 30 participants at a time doing physical exercise. Additionally, it seems to be common knowledge amongst the low threshold services that mental illness is taboo among Somalis: "I have noticed that the GPs can be very desperate when they do not get their patients to the DPS. Mental health is taboo. . ..there is a lot of pain and they take painkillers, are tired, sad and lives a silent life . . . I think many are too ill to be in the HCL." (HLC, female, age 45). There is also an impression in the Healthy Life Centers that the GPs can be a little desperate when referring the patient to them: "We have experienced situations where a GP has sent patients to us who should not be here because their diagnoses are too complicated. Once we got one in with schizophrenia, and we told the GP, but he replied: "What then shall I do with this patient?" (HLC, female, age 45). Another informant said: "There are probably many here who should have been at DPS or RPH, but many will not get admitted, and it's an eternal dilemma as to who's getting help or not." (HLC leader, female, age 60).

The fact that the focus is placed on somatic conditions, encourages women to use these services. There are often psychologists employed; as well as nurses, physiotherapists and nutrition specialists. The timeframe for communication is wider than in other subsystems but is more similar to the timeframe within Activity Houses. Life stories comes up that can be discussed and analyzed together with others while participants walk and share activities." The other day, during a trip in the woods, a Somali woman told us that her whole family had been beheaded, and she had watched it all. She was much traumatized." (HLC, female, age 45).

The subsystems at the levels above and other subsystems at the municipal level seem to redirect their patients to HCL. As the main focus lies on physical health, it becomes an acceptable service for user groups who consider mental health taboo. These centers meet the demand for help, but do not provide the professional psychological treatment provided at the specialist level. Psychological issues, as well as FGM/C, are still issues that can remain silenced, even though there is an awareness among those employed that these issues may underlie some physical conditions and pain. "We have never touched upon the topic of FGM/C. . . I think depression has had an impact, and they need mental help and treatment." (HCL, female, age 31).

However, the deference norms on both sides, attitudes and high standards of referrals represent a barrier that creates a block in the system that impedes health workers, who will not invade users' privacy, and patients, who wants to protect their privacy, from addressing FGM/C and mental health.

## Conclusion

The health services that used to be organized on only two levels have developed into a three-level system, with a low entry threshold at the bottom. The low threshold institutions are not formally recognized as belonging to a third level, but there is no requirement for a formal referral and no obligatory feedback to the GP, like the levels at the top. This lack of feedback justifies defining the health services at this level as something different from the traditional system of the levels above.

The first entry of the patient into the system is through the GP consultancy room. The fact that there are new forms of sufferings, different languages and different codes of conduct and that new forms of deference are entering consultancy rooms makes GPs and health workers' role of interpreting and diagnosing mental health more challenging. Sensitive issues and different deference codes thereby demands a lot more time for mental and private health issues to become visible for the GP. In addition, it is not certain that the women suffering from physical and mental health consequences due to FGM/C, see the link of their suffering due to the FGM/C procedure many years ago. To discover both the health problem and the link, may

require time within the consultancies in order to dig deep into these sensitive issues. In addition there is also a need for competence by the health workers to see these links. Extra time and effort to reveal links and suffering is not necessarily possible given the current system organization, so these issues become silenced.

Presently, FGM/C is not a topic that patients generally bring up with their doctors, neither their GP nor at the DPS in the second line services nor at other health services at the bottom. This finding that FGM/C is silenced due to different factors, confirms the findings of the reviews of health services [22].

There are some obvious reasons for the "crisis of the GP scheme," as subsystems at upper levels try to protect their own subsystems in order to achieve sustainability and balance. The protective factors around subsystems may be professionally justified by standard requirements, but create extra work and confusion for GPs when patients in need of treatment are sent back or referral letters are not taken seriously or must be rewritten. The waiting lists and the high referral standards have constructed a protective boundary around the subsystems, making it difficult for many SSA patients to enter levels above.

Deference attitudes stem from cultural values of showing respect for the patient, but these attitudes, taken too far, can create both indifference to health issues and a boundary that silences issues. These factors may impede a timely diagnostic assessment of the effect of FGM/C. The attitude that only one issue at a time should be addressed in the consultancy room can prevent health problems from surfacing. There are also other silencing mechanisms and attitudes related to mental health that can have a "cooling effect on the temperature of the system" and reduce the speed of referrals and treatment. There has been expressions from specialists that suggest a lack of trust that existing methods of treatment can help SSA women The combination of these realities and attitudes may silence FGM/C as well as mental health issues in general and prevent their treatment.

We can call these arguments "the attitudes of the system" that in a socio-cybernetic analysis represent "the thermometer" that regulates the flow of communication and patients, slows down the movement of patients and redirects them. A cybernetic analysis has provided tools for studying a complex mental health system with subsystems that communicates and interacts with each other. With this analysis we have been able to see where there are protective borders. We can also see how the subsystems are integrated with each other and where it lacks integration, and what the thermometer of the system contains. Silencing and ignoring health issues due to a narrow time frame and deference rules, as well as setting referral standards at a very high level is part of the thermometer that balances the system and protects subsystems from overflow and overheating, but these practices can also create health risks for the patients.

When therapist think that "these patients do not fit in any of our treatment groups" or that these patients cannot take advantage of health services, the flow of these patients within the system will be redirected, sent back to the GP, sent to the bottom line, or cease altogether. Due to these boundaries and attitudes, many needy patients thereby are pushed away from the second line services, psychologists of agreement, the DPS, and also from some of the services at the municipal level, like the RPH and the Activity Houses, which have a strong focus on mental health therapy. The places where this study found many SSA women with mental health issues who suffered potential consequences of FGM/C were at the Healthy Life Centers and the Health Forum for Women. The Healthy Life Centers focus on workouts and physical training, and here we find groups of Somalian and other SSA women suffering from depression and potential mental consequences of FGM/C.

When meeting many obstacles from other subservices, the GP may give up or even learn to ignore issues that are defined as minor, difficult or not life-threatening. The belief that SSA women themselves do not feel that FGM/C is an important issue to address, makes it easier to

turn a blind eye to the problem. Many GPs, however, care a lot for their patients, but the combined attitude of the system (that these patients do not fit in, that they are sensitive, and that their patient-doctor relationships are precarious) will continue to silence the topic.

The attitudes of the system, or the "the thermometer," is based on caring ethics, deference and respect. However, these attitudes can contribute to the fact that certain mental health issues and physical and psychological consequences of FGM/C are under-investigated, under-referred, and under-treated and silenced problems within the health services. It also leads to the consequence that SSA women do not get sufficient professional psychological help at the specialist level. Instead they are directed away from the professionally based mental health services over to more physically and exercise oriented services at the municipal level of the health system, which the women can benefit from, but may not all the same be sufficient to their needs.

To conclude this study we can say that the crises of the GP-scheme can partly be explained by the protective borders of the sub-systems above, requiring a better integration of the subsystem at the specialist level with the GP scheme. Secondly, it is important to provide competence on both physical and mental health consequences of FGM/C to health workers at all levels. In addition there is a need to provide knowledge on the possible link between certain forms of sufferings and the FGM/C procedure. SSA women also need health system competence. They need to know where the available services are to be found, especially at the municipal level. These low threshold services needs to be further strengthened, integrated and expanded, as they combine mental treatment with exercise and gives opportunities for communication in a culturally sensitive way that can be accepted and appreciated by the women.

There is also a need for more research on the health care system itself and its relationship with new user groups. Does immigrant women get equal access to health care? Is there enough competence on FGM/C in the different subsystems of the health care system? How are subsystems integrated with each other, and how does the referral system work in different countries? Socio-cybernetic system analysis can provide useful research tools to investigate the way the health system work as a total system, and answer some of these questions relating to the systems adaptation to new user groups, and particularly to those women who suffer health consequences of FGM/C.

## Limitation and strength of the study

The study would have benefitted from more interviews with informants at the Regional Psychiatric Centers (DPS). However, it was difficult to get informants, keeping the number of interviews from these institutions small, which has created a limitation of the study. Likewise, there is a lack of interviews from the Psychologists with agreement (Avtalepsykologer), that could have explained their inside perspective on the factors that we have described as protective borders and lack of feedback as experienced by the GPs and other health workers. The rationale that these psychologists have at the specialist level for assessing and dismissing referrals would thereby have been clearer. The presentations of the services at the specialist level is thereby an "outside" perspective to a great extent, seen from another subsystem position within the total system, which is a limitation of the study. However, informants situated at different positions in the total system, reflecting on the connections and communication with other subsystems, is a strength of the study. These multi-situated perspectives also gives credibility and substance to the analysis of the total system.

## Supporting information

**S1 Table. Informants and interviews.**
(DOCX)

**S1 Appendix. A semi-structured interview-guide for health-personnel at different levels and institutions.**
(DOCX)

## Author Contributions

**Conceptualization:** Inger-Lise Lien, Cecilie Knagenhjelm Hertzberg.

**Funding acquisition:** Inger-Lise Lien.

**Investigation:** Inger-Lise Lien, Cecilie Knagenhjelm Hertzberg.

**Methodology:** Inger-Lise Lien.

**Project administration:** Inger-Lise Lien.

**Supervision:** Inger-Lise Lien.

**Validation:** Inger-Lise Lien.

**Visualization:** Inger-Lise Lien.

**Writing – original draft:** Inger-Lise Lien, Cecilie Knagenhjelm Hertzberg.

**Writing – review & editing:** Inger-Lise Lien.

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
