## [Decision Letter · Decision Letter 0]

6 Mar 2020

PONE-D-20-00924

A system analysis of the mental health services in Norway available to women suffering from mental health problems in general and in particular due to female genital mutilation/cutting

PLOS ONE

Dear Dr. Researcher 1 (Research Professor) Lien,

Thank you for submitting your manuscript to PLOS ONE. After careful consideration, we feel that it has merit but does not fully meet PLOS ONE’s publication criteria as it currently stands. Therefore, we invite you to submit a revised version of the manuscript that addresses the points raised during the review process.

Please see reviewer comments below. In your revised submission, please respond to each item discussed by the reviewers.

We would appreciate receiving your revised manuscript by Apr 20 2020 11:59PM. To enhance the reproducibility of your results, we recommend that if applicable you deposit your laboratory protocols in protocols.io, where a protocol can be assigned its own identifier (DOI) such that it can be cited independently in the future. For instructions see: http://journals.plos.org/plosone/s/submission-guidelines#loc-laboratory-protocols

We look forward to receiving your revised manuscript.

Kind regards,

Amy Michelle DeBaets, PhD

Academic Editor

PLOS ONE

Journal Requirements:

2. Please address the following:

- Please ensure you have thoroughly discussed any potential limitations of this study within the Discussion section.

- Please include additional information regarding the survey or questionnaire used in the study and ensure that you have provided sufficient details that others could replicate the analyses. For instance, if you developed a questionnaire as part of this study and it is not under a copyright more restrictive than CC-BY, please include a copy, in both the original language and English, as Supporting Information.

3. Your ethics statement must appear in the Methods section of your manuscript. If your ethics statement is written in any section besides the Methods, please move it to the Methods section and delete it from any other section. Please also ensure that your ethics statement is included in your manuscript, as the ethics section of your online submission will not be published alongside your manuscript.

4. Please ensure that you refer to Figure 1 in your text as, if accepted, production will need this reference to link the reader to the figure.

Reviewers' comments:

Reviewer's Responses to Questions

**Comments to the Author**

1. Is the manuscript technically sound, and do the data support the conclusions?

Reviewer #1: Partly

Reviewer #2: Yes

2. Has the statistical analysis been performed appropriately and rigorously? 

Reviewer #1: No

Reviewer #2: N/A

3. Have the authors made all data underlying the findings in their manuscript fully available?

Reviewer #1: Yes

Reviewer #2: No

4. Is the manuscript presented in an intelligible fashion and written in standard English?

Reviewer #1: Yes

Reviewer #2: Yes

5. Review Comments to the Author

Reviewer #1: Generally, the manuscript is written acceptable and “the mental health services in Norway” can be a good experience for other countries. Please see my comments:

1- The title is long. I suggest you delete this section: "available to women suffering from mental health problems in general and in particular due to female genital mutilation/cutting"

2- Lines 66 to 70 require references. Please see below article:

Effect of female genital mutilation on mental health: a case–control study

https://www.tandfonline.com/doi/abs/10.1080/13625187.2019.1709815

3- Please write more detail about sampling (e.g. maximum variation or other type of sampling and why?)

4- How many of informants were interviewed twice?

5- Why "women suffering from mental health problems and FGM/C" are not in interview list? And how do you analyze "the mental health services" without considering the customer's viewpoint?

6- This sentence has been repeated twice: “Some informants were interviewed several times and were used as key informants” (lines 162 and 179).

7- Please add the methods of analysis and coding of interviews.

8- This is part of the findings (not a separate section) “A system of mental health and its borders and entries”

Reviewer #2: Thank you for the opportunity to review this paper. It is on an important topic and, I believe, it does make an important contribution to the general field.

There are aspects of the paper however, as it is currently written, that I find a bit confusing and/or problematic. I outline these below.

In addition, I think the paper would benefit from more proof reading/copy editing with regards to English language.

GENERAL COMMENTS

Two key assumptions need to be examined more and explained. The first is that because women are not referred for psychological help, specifically for FGC related issues, the system has failed them. However, there is not any strong evidence presented that women with FGC actually want or need psychological help specifically for FGC-related issues. By saying this, I am not denying that they may need help for FGC-related issues or for other traumatic issues or other psychological needs, but I am suggesting that the authors need to create a much stronger argument to justify their statements that women from Somalia/Eritrea need psychological help specifically for FGC? Where is the evidence from women with FGC showing that they need FGC-specific psychological support? This evidence needs to be cited. In addition, it is interesting that the data show that the professionals in the study who currently work with these women do not seem to state that they need it. Nor do they give personal examples of any experiences with women who want psychological help specific to FGC. Hence, it currently feels like the argument in the paper is proceeding based on an assumption that FGC-specific psychological help is an important issue. I think this assumption needs to be more thoroughly evidenced, problematised and discussed – both in the background section and then again in the discussion. It seems that the paper has started off trying to prove that there is a problem, without actually really showing that there is. The evidence presented shows that there certainly seem to be generic problems with help-seeking and communication and accessing treatment for mental health issues amongst immigrant populations in general. But whether or not these are specifically linked to FGC is rather unclear.

The second assumption is the implied criticism of the system because women from SSA access help through the health life centers. In fact, the data seems to suggest that these centers are meeting needs in a very culturally appropriate way – by providing a way into care through a focus on physical health, but then by providing access to other services and, importantly, a community where women feel safe and able to talk. Yet, the authors are referring to them as being at the ‘bottom’ of the system. On what basis is this criticism made? Perhaps in fact, more centers such as this need to be set up and supported?

ABSTRACT

I think sub-headings are required in the abstract. The abstract implies that the focus on the paper is with data from a sub-sample of GPs. However this is not the case. Hence, I think the abstract needs to be re-written a little to reflect the actual paper.

BACKGROUND

Lines 84-88 (explaining levels of mental health amongst Somalis and Eritreans) – uses colloquial language (“described as a mystery”) – please re-word.

A key issue in this section is that it is slightly unclear why the authors have chosen to focus upon mental health when they have described that levels of mental health among Somali/Erirean women appear to be similar to the general population (i.e. the case for the focus on mental health needs to be stronger)?

In several places, the term “i.e.” seems to appear inappropriately (lines 93, 94)

The sentences in the paragraph in lines 93-101 seems rather poorly constructed. I would suggest re-writing. Does it need a sub-heading “Research Aim”?

The background provides a brief description of the organisation of the GP system in Norway. I would suggest that the whole ‘mental health’ system needs to be more clearly and properly explained.

The section on the health system as a ‘socio-cybernetic system’ needs a bit more elaboration, including more examples of how and when and why it has been used in healthcare. I think there also needs to be more explanation as to why this approach was chosen in contrast to other possible approaches? What new insights can be gained by using this approach?

METHOD

There needs to be a discussion relating to reflexivity

How was data analysis approached?

What measures were adopted to enhance credibility and dependability of the research?

A SYSTEM OF MENTAL HEALTH AND ITS BORDERS AND ENTRIES

The section in lines 231-242 about Somali’s health seeking behaviour would be better placed in the Background section as part of the rationale for doing the study.

The finding about the GPs referral letters not being accepted is interesting but needs more explanation. As a non-Norwegian, I can’t understand why another service would reject a referral. Did the GPs give any examples of women from SSA who they had referred for depression but whose referral had been rejected?

The section on ‘health centers’ (lines 325-339) suggest that midwives see more patients with FGC than GPs (or discuss it more openly). The authors note that midwives do not have referral right to send a patient for psychological care. However, the paper does not make it clear whether or not the midwives would like to have this right or whether they feel they need it. What did the midwives have to say about women with FGC and depression?

CONCLUSION

This section should be re-titled ‘Discussion’ and should relate the findings to the wider literature and policy. The section needs to include recommendations for policy and future research? In addition, it should include a discussion on the value of having utilised a socio-cybernetic system approach.

6. PLOS authors have the option to publish the peer review history of their article (what does this mean?). If published, this will include your full peer review and any attached files.

Reviewer #1: No

Reviewer #2: Yes: Dr Catrin Evans

---

## [Author Response · Author response to Decision Letter 0]

18 Apr 2020

To Reviewer #1: Thank you very much for the review. We have point by point tried to improve the manuscript according to very useful the suggestions. The changes is explained in the Respons to Reviewer letter.

To Reviewer # 2: Thank you very much for the response. We have tried our best to improve the article according to your very good advice, and moved text according to suggestions from you. So thank you again.

---

## [Decision Letter · Decision Letter 1]

12 Oct 2020

A system analysis of the mental health services in Norway and its availability to women with female genital mutilation

PONE-D-20-00924R1

Dear Dr. Lien,

We’re pleased to inform you that your manuscript has been judged scientifically suitable for publication and will be formally accepted for publication once it meets all outstanding technical requirements.

Kind regards,

Juliet Kiguli, MA, PhD

Academic Editor

PLOS ONE

Additional Editor Comments (optional):

This manuscript has addressed all comments and can now be accepted. Thanks!

Reviewers' comments:

Reviewer's Responses to Questions

**Comments to the Author**

1. If the authors have adequately addressed your comments raised in a previous round of review and you feel that this manuscript is now acceptable for publication, you may indicate that here to bypass the “Comments to the Author” section, enter your conflict of interest statement in the “Confidential to Editor” section, and submit your "Accept" recommendation.

Reviewer #1: All comments have been addressed

Reviewer #2: All comments have been addressed

2. Is the manuscript technically sound, and do the data support the conclusions?

Reviewer #1: Yes

Reviewer #2: Yes

3. Has the statistical analysis been performed appropriately and rigorously? 

Reviewer #1: Yes

Reviewer #2: N/A

4. Have the authors made all data underlying the findings in their manuscript fully available?

Reviewer #1: Yes

Reviewer #2: No

5. Is the manuscript presented in an intelligible fashion and written in standard English?

Reviewer #1: Yes

Reviewer #2: Yes

6. Review Comments to the Author

Reviewer #1: Dear authors,

Thank you for your very careful and extensive revision. All comments have been addressed.

Reviewer #2: Thank you. You have addressed all my queries. This is an interesting study and I look forward to seeing the subsequent papers

7. PLOS authors have the option to publish the peer review history of their article (what does this mean?). If published, this will include your full peer review and any attached files.

Reviewer #1: No

Reviewer #2: No

---

## [Editor Report · Acceptance letter]

22 Oct 2020

PONE-D-20-00924R1 

A system analysis of the mental health services in Norway and its availability to women with female genital mutilation 

Dear Dr. Lien:

I'm pleased to inform you that your manuscript has been deemed suitable for publication in PLOS ONE. Congratulations! Your manuscript is now with our production department. 

Kind regards, 

on behalf of

Dr. Juliet Kiguli 

Academic Editor

PLOS ONE